# Mixed-methods analysis of select issues reported in the 2016 World Health Organization verbal autopsy questionnaire

Erin Nichols[1]*, Kristen Pettrone[1,2], Brent Vickers[3], Hermon Gebrehiwet[4], Clarissa Surek-Clark[5], Jordana Leitao[6], Agbessi Amouzou[7], Dianna M. Blau[8], Debbie Bradshaw[9], El Marnissi Abdelilah[10], Pamela Groenewald[9], Brian Munkombwe[1], Chomba Mwango[11], F. Sam Notzon[12], Steve Biko Odhiambo[13], Paul Scanlon[3]

1 Division of Vital Statistics, National Center for Health Statistics, Centers for Disease Control and Prevent, Hyattsville, Maryland, United States of America, 2 Epidemic Intelligence Service, Centers for Disease Control and Prevent, Atlanta, Georgia, United States of America, 3 Collaborating Center for Questionnaire Design and Evaluation Research, Division of Research and Methodology, National Center for Health Statistics, Centers for Disease Control and Prevention, Hyattsville, Maryland, United States of America, 4 School of Nursing and Health Sciences, Public Health Program, Capella University, Minneapolis, Minnesota, United States of America, 5 Departments of English and Sociology, College of Arts and Sciences, The Ohio State University, Columbus, Ohio, United States of America, 6 World Health Organization, Geneva, Switzerland, 7 Institute for International Programs, Department of International Health, Johns Hopkins Bloomberg School of Public Health, Baltimore, Maryland, United States of America, 8 Child Health and Mortality Prevention Surveillance (CHAMPS), Center for Global Health, Centers for Disease Control and Prevention, Atlanta, Georgia, United States of America, 9 Burden of Disease Research Unit, South African Medical Research Council, Cape Town, Western Cape, South Africa, 10 Planning and Studies Division, Directorate of Planning and Financial Resources, Ministry of Health, Rabat, Morocco, 11 Bloomberg Data for Health Initiative, Lusaka, Zambia, 12 CDC Foundation, Atlanta, Georgia, United States of America, 13 Kenya Medical Research Institute, Centre for Global Health Research, Health and Demographic Surveillance System, Kisumu, Kenya

* igd1@cdc.gov

**Data Availability Statement:** We are only able to consider data sharing by request. While the data are de-identified, some of the data are sourced

## Abstract

### Background

Use of a standardized verbal autopsy (VA) questionnaire, such as the World Health Organization (WHO) instrument, can improve the consistency and reliability of the data it collects. Systematically revising a questionnaire, however, requires evidence about the performance of its questions. The purpose of this investigation was to use a mixed methods approach to evaluate the performance of questions related to 14 previously reported issues in the 2016 version of the WHO questionnaire, where there were concerns of potential confusion, redundancy, or inability of the respondent to answer the question. The results from this mixed methods analysis are discussed across common themes that may have contributed to the underperformance of questions and have been compiled to inform decisions around the revision of the current VA instrument.

### Methods

Quantitative analysis of 19,150 VAs for neonates, children, and adults from five project teams implementing VAs predominately in Sub-Saharan Africa included frequency

from sensitive data that contribute to vital statistics. The data have been provided by teams for this analysis with the agreement that the data will only be used by the agreed collaborators for the purpose of contributing to the improvement of the WHO verbal autopsy questionnaire. Data contributors agreed to the publication of aggregate findings and are co-authors of the paper. We can address any requests for the data individually; requests can be directed to the corresponding author or to Daniel Cobos Muñoz, MD MSc PhD, Project Leader, Health Systems and Policies Research Group, Epidemiology and Public Health Department, Swiss Tropical and Public Health Institute, daniel.cobos@swisstph.ch.

**Funding:** The authors received no specific funding for the secondary analysis presented in this manuscript; their time for the analysis and drafting of this manuscript was supported in-kind from their respective institutions. While some co-authors and their institutions were involved through separate projects in collecting the original data, no funders had any role in the secondary analysis, decision to publish, or preparation of the manuscript.

**Competing interests:** The authors have declared that no competing interests exist.

distributions and cross-tabulations to evaluate response patterns among related questions. The association of respondent characteristics and response patterns was evaluated using prevalence ratios. Qualitative analysis included results from cognitive interviewing, an approach that provides a detailed understanding of the meanings and processes that respondents use to answer interview questions. Cognitive interviews were conducted among 149 participants in Morocco and Zambia. Findings from the qualitative and quantitative analyses were triangulated to identify common themes.

## Results

Four broad themes contributing to the underperformance or redundancy within the instrument were identified: question sequence, overlap within the question series, questions outside the frame of reference of the respondent, and questions needing clarification. The series of questions associated with one of the 14 identified issues (the series of questions on injuries) related to question sequence; seven (tobacco use, sores, breast swelling, abdominal problem, vomiting, vaccination, and baby size) demonstrated similar response patterns among questions within each series capturing overlapping information. Respondent characteristics, including relationship to the deceased and whether or not the respondent lived with the deceased, were associated with differing frequencies of non-substantive responses in three question series (female health related issues, tobacco use, and baby size). An inconsistent understanding of related constructs was observed between questions related to sores/ulcers, birth weight/baby size, and diagnosis of dementia/presence of mental confusion. An incorrect association of the intended construct with that which was interpreted by the respondent was observed in the medical diagnosis question series.

## Conclusions

In this mixed methods analysis, we identified series of questions which could be shortened through elimination of redundancy, series of questions requiring clarification due to unclear constructs, and the impact of respondent characteristics on the quality of responses. These changes can lead to a better understanding of the question constructs by the respondents, increase the acceptance of the tool, and improve the overall accuracy of the VA instrument.

## Introduction

Verbal autopsy (VA) is a method for estimating population-level cause of death information for mortality surveillance purposes, in the absence of physician certification of cause of death or full autopsy as is the case in most low-to-middle-income countries [1, 2]. VA involves a structured interview conducted by a trained interviewer, in which family members or caregivers familiar with the deceased provide information about the signs, symptoms, medical history and circumstances experienced by the deceased at and around the time of death. From this information, a cause of death is determined. Cause of death determination from VA can be done using physician review, an expert-derived algorithm, or a computer-coded algorithm.

Over the last 15 years, there has been a push to develop a standardized VA tool in an effort to improve consistency and comparability between countries and to address concerns about the validity of instruments and the comparability of data [3]. In 2007, the World Health

Organization (WHO) introduced the first international technical standards and guidelines for VA [4]. The instrument was updated in 2012, 2014 and 2016; the current version is the 2016 WHO VA instrument, which is now used by more than 20 countries and includes questions addressing COVID-19 [5, 6].

When updating or revising a questionnaire, such as a VA tool, the overall objective is to create a shorter and more practical instrument while increasing the value and reliability of its individual questions [7–10]. This can increase the acceptability of VA by respondents and communities, decrease question non-response, and improve the validity and utility of the VA process [11]. Systematically revising a questionnaire, however, requires evidence about the performance its questions. One robust approach to uncovering this evidence is to use a mixed method question evaluation approach.

Mixed methods research is a process which integrates information from both qualitative and quantitative sources, leveraging the strengths of each epistemology, to produce a better understanding of findings by compensating for each method's limitations [12, 13]. Grounded within logical empiricism, quantitative studies examine associations between key factors and outcomes in accordance with observed empirical laws but are subject to bias that can be controlled to some extent by research design. Qualitative studies, grounded within constructivism, use an interpretive process to contextualize findings. Used together, qualitative findings can help explain associations observed in quantitative findings [13]. Greene et al. have described several uses of a mixed methods approach including enhancement or clarification of results from one method with another method, the expansion of an investigation using different methods, and a discovery of new perspectives or contradictions in one method using the results of another [14]. Benitez-Baena and Padilla describe a series of approaches that can be used to mix qualitative and quantitative methods in the evaluation of survey instruments, including the use of experimental designs, within-survey probes, and multi-mode cognitive interviewing, an approach that provides a detailed understanding of the meanings and processes that respondents use to answer interview questions [15]. Others have successfully combined cognitive interviewing with psychometric approaches like Item Response Theory (IRT) to reduce and validate scales or cognitive interviews with probes allowing for greater extrapolation of cognitive findings [16–18].

The purpose of this investigation was to use a mixed methods approach triangulating cognitive interviewing and questionnaire performance findings to evaluate the performance of select questions in the 2016 version of the WHO questionnaire, as part of a broader questionnaire revision effort [1]. Feedback from field teams who use the WHO standard VA questionnaire was collected via a public facing GitHub platform commencing in 2017 [19]. The platform was established for VA users to report and track issues identified during use of the standard questionnaire; the types of issues reported via this platform vary from minor issues in the electronic programming of the questionnaire or suggestions to add hints for clarity to more significant issues including incorrect skip patterns or confusion about questions that are likely to impact the quality of responses. A review of feedback in 2019 identified, 14 problematic issues for which solutions could be well-informed through a mixed methods approach (Table 1). These issues relate to the how questions impact the interview process (for example, causing repetition or unnecessarily lengthening the interview) and quality of responses (for example, where there is lack of clarity on what a question is asking or the question is not answerable by the respondent). VA results from multiple countries provided the quantitative data for this evaluation, while qualitative findings were derived from cognitive interviews of VA respondents in two countries. The results from this mixed methods analysis have identified common themes that may have contributed to the underperformance of questions and are being used to inform decisions around the revision of the VA instrument. More broadly, the application of this methodology can be used in the evaluation of other cross-cultural survey instruments.

**Table 1. Series of questions identified for review from feedback from field teams and description of associated issue.**

| Series | Description of Issue |
|---|---|
| 1. Tobacco Use | Can the series be shortened? |
| | Can the respondent provide meaningful responses when asked about the number and frequency of tobacco products used? |
| 2. Swallowing | What is the consistency of responses to the questions of "pain" and "difficulty"? |
| | Are respondents able to differentiate the constructs of "pain" and "difficulty"? Can or should one question be eliminated? |
| 3. Sores and Ulcers | This question series asks multiple questions about similar though not identical constructs. Are the constructs clearly understood? |
| | Can the question series be shortened? |
| 4. Swelling, Lump, Ulcers, Pits in the Breast | There is potential confusion between the constructs of swelling or lump in the breast and ulcers (pits) in the breast. Are participants able to answer these questions? |
| | Are both questions needed? |
| | What are the response patterns by respondent characteristics? |
| | Are the response patterns different in those with greater familiarity with the deceased? |
| 5. Other Female Health-Related Questions | Are the response patterns to these gendered questions different in those with greater familiarity with the deceased? |
| 6. Medical Diagnosis Questions | Measurement or response error is more likely with questions on diagnosis than with questions on symptoms. Are there response patterns to the symptoms questions that correspond to the medical diagnosis that provide an evidence of response error? What is the respondent understanding and interpretation of the medical diagnosis questions? |
| 7. Vaccinations | The question "Select EPI Vaccines Done" requires the interviewer to know what the complete vaccine schedule is for their country and to assess the vaccination card for completion. With this complexity, there is much room for error. Also, documentation of vaccine status is required for a response to one question in the series; a concern has been reported that for many respondents, this documentation may not be available, because it was thrown away, buried with the child, or otherwise lost. How has this question performed? Can it be simplified? |
| 8. Injury Questions | Is the full VA required for those deceased who clearly died of an injury? |
| 9. Urine | What is the consistency between a "Yes" response to the first order question and a "Yes" response to the more detailed, second order questions? Inconsistencies would flag potential false positives; respondent may not know what urine problems are (e.g., blood in pee)? |
| | For important questions, is it better to ask the specific construct of interest directly and not screen out based on the response to the first order question? |
| 10. Abdominal Problem | There is potential for redundancy and/or inconsistency across this series of questions. Can we shorten this series in any way? |
| 11. Lumps | Are the response patterns different in those with greater familiarity with the deceased? |
| 12. Vomiting | The questions "Did (s)he vomit?" and "To clarify: Did (s)he vomit in the week preceding the death?" are both asked of all respondents. Can one question be eliminated? |
| | The question "How long before death did s(h)e vomit?" required clarification. Does it refer to the duration or timing of the vomiting? |
| 13. Violence | There is a concern of under-reporting of suicide for children. |
| | What is the consistency in responses to violence and self-inflicted injury for children? |
| 14. Baby Size | What is the consistency in responses to the series of questions about size and weight? |
| | What is the frequency and plausibility of the responses to the reported birth weight? |

## Methods

### Quantitative data collection

Data from VA questionnaires were compiled into two datasets for this analysis—the primary dataset, which included VA interview data from five project teams, and the reference dataset, which included the VA interview data in the primary dataset as well as information on the deceased's cause of death, as determined by physician-certified VA, which was available from one project team. The primary dataset consisted of de-identified, aggregated VA results using the 2016 WHO VA questionnaire. Project teams that had completed at least 1,000 VAs conducted by field interviewers (typically, though not always, a community health employee) using the 2016 WHO VA questionnaire and who agreed to contribute their data for the exercise submitted data to the analysis team. The following countries and sources contributed to the dataset: Zambia (VAs conducted by the Department of National Registration, Passports, and Citizenship for community deaths "brought in dead" to two mortuaries in Lusaka, Zambia); South Africa (as described below), Kenya (VAs from the Kenya Medical Research Institute/U.S. Centers for Disease Control and Prevention (CDC) Health and Demographic Surveillance Site in Western Kenya), CHAMPS (Child Health and Mortality Prevention Surveillance) and COMSA(Countrywide Mortality Surveillance for Action)-Mozambique. The CHAMPS network focuses on mortality surveillance in children under 5 years of age in sub-Saharan Africa and South Asia [20]. COMSA-Mozambique is a surveillance program in Mozambique that produces and makes publicly available continuous annual data on mortality and cause of death at national and subnational levels within the country [21].

The reference dataset included the VA interview data as in the primary dataset in addition to cause of death information determined by physician review of the VA interview (PCVA or physician certified VA) from the South Africa National Cause of Death Validation Study [22]. With relevance to the reference dataset, there was no difference in how the VA interview was administered for the data in the reference dataset compared to the primary dataset. The reference dataset contributed to analysis for three issues, including the medical diagnosis questions, injury, and violence. De-identified data were transferred by country teams and stored via a secure share-file system with access restricted only to data contributors and analysts.

The primary dataset for the quantitative analysis included 19,150 verbal autopsies: 13,736 adults (persons aged 12 years and above), 2,916 children (aged 4 weeks to 11 years) and 2,498 neonates (under 4 weeks old); of these, 10,280 were males and 8,870 were females. The reference dataset, which included cause of death information, contained 5,389 verbal autopsies: 102 neonates, 187 children and 5,100 adults; 2,579 were female and 2,810 were male. Cause of death in the reference dataset was determined using the physician-certified VA [22].

### Qualitative data collection

Cognitive interviewing is a qualitative method whose purpose is to evaluate survey questionnaires and determine which constructs the questionnaires' items capture. The primary benefit of cognitive interviewing over non-qualitative evaluation methods is that it provides rich, contextual data into how respondents interpret questions, apply their lived experiences to their responses, and formulate responses to survey items based on those interpretations and experiences [23]. Thus, cognitive interviewing data allows researchers and survey designers to understand whether or not a question is capturing the specific social constructs they originally wanted and gives insight into what design changes are needed to advance the survey's overall goal. Cognitive interviews were conducted in Zambia and Morocco in 2019. These sites were selected based on their readiness, given that the VA process had been well established and the

field teams were interested in evaluating the performance of the interview process using cognitive interviewing. Staff from the U.S. CDC's National Center for Health Statistics (NCHS) Collaborating Center for Questionnaire Design and Evaluation Research (CCQDER) trained interviewers selected from the local communities on cognitive interviewing during a week-long, on-site training. Following the training in English, the interviewers recruited cognitive interviewing participants among the VA respondents, such that the selected VA respondents were also the cognitive interviewing participants; no data in the VA interview were changed as a result of cognitive interviewing discussions. In Zambia, recruitment occurred at one of three hospitals in the Lusaka area, whereas in Morocco respondents were recruited from Ministry of Interior offices in the Rabat area where relatives came to certify deaths. Written and verbal informed consent was obtained from all respondents. A total of 149 semi-structured interviews (n = 84 in Morocco and n = 65 in Zambia) were conducted in the native language of the respondents across the two sites via a purposive sample, with adult respondents recruited in order to examine all three of the VA questionnaires (adult, child, and neonate); as a result, 68 respondents received the adult questionnaire (45 in Morocco, 23 in Zambia), 43 received the child questionnaire (20 in Morocco, 23 in Zambia), and 38 received the neonatal questionnaire (19 in Morocco and 19 in Zambia). The interview structure consisted of respondents first answering the VA questions and then answering a series of follow-up probe questions that reveal what respondents were thinking and their rationale for that specific response. While there was a selection of questions that all the interviewers probed on, based on problematic areas identified during VA implementation in Morocco and Zambia, they were also given free rein to probe on any other questions they thought the respondents did not understand or questions they thought might have elicited response errors during the VA interview. Most interviews lasted approximately 30–60 minutes beyond the normal VA interview. Cognitive interviewers recorded their initial interview notes in the language of their preference, then entered their notes in English into CDC's Q-Notes software, which is a qualitative analysis program designed specifically for the storage and analysis of data from cognitive interviews [24]. Interviews were conducted over a period of six months in Zambia (January-June 2019) and one year in Morocco (February 2019- February 2020). CCQDER researchers monitored data collection and quality via Q-Notes and communicated with the field teams when necessary to provide direction and assistance. Once all interviews were complete, CCQDER staff analyzed the interview notes and summarized the findings using an iterative five-step synthesis and reduction process: conducting interviews, producing summaries, comparing across respondents, comparing across subgroups of respondents, and reaching conclusions [23, 25, 26]. As is common across cognitive interviewing studies, this analysis process uncovered the patterns of interpretation used by the respondents to comprehend, judge, and respond to the various survey items under study.

This activity was reviewed by CDC and conducted consistent with applicable federal law and CDC policy as an institutional review board–exempt public health surveillance evaluation. Approval for cognitive interviewing and qualitative data collection was obtained from the University of Zambia Biomedical Research Ethics Committee and the Mohammed V University Comité d'Éthique pour la Recherche Biomédical de Rabat.

## Analysis

This investigation included mixed methods analysis of secondary data collected using the 2016 WHO VA questionnaire together with cognitive interviewing results. All data were de-identified by the contributing teams prior to submission for analysis. Quantitative results provided information on performance of the series of questions related to the 14 selected issues,

including response pattern analysis and the association of respondent characteristics with response patterns. Frequency distributions and cross tabulations of quantitative data were run to compare response patterns among related questions. The impact of respondent characteristics on ability to provide substantive responses was evaluated by calculating prevalence ratios, 95% confidence intervals and $p$-values using the chi-squared test with $p<0.05$ considered significant. "Yes" and "No" were classified as substantive responses while "Don't Know" and "Refused" were classified as non-substantive responses. For respondent characteristics, close family members were categorized as being a sister, parent, child, or spouse. The analysis was run using the SAS v.9.4 statistical software system (SAS Institute, Cary, NC, USA). Qualitative cognitive interviewing results for series of questions related to the 14 select issues provided insight on the ways in which the question was interpreted by various groups of respondents, the processes that respondents utilized to formulate a response as well as any difficulties that respondents might have experienced when attempting to answer the question. Qualitative and quantitative results were triangulated iteratively and integrated into summary findings. Results and summary findings from each issue were then analyzed to identify commonalities or themes that may have contributed to the underperformance of the question and to make recommendations for improvement.

## Results

From the application of the mixed methods analysis to the 14 issues (Table 1) flagged by end-users of the 2016 WHO standard questionnaire, we identified four broad themes contributing to the underperformance of or redundancy within the instrument: question sequence, overlap or redundancy within the question series, questions outside the frame of reference of the respondent, and questions or concepts needing clarification. Each of these constructs will be discussed in more detail with select examples drawn from the 14 pre-identified issues to demonstrate the application of various analysis types. A full description and analysis of each of the 14 issues is available in S1 File.

### Question sequence

Question sequence addresses the ordering of questions within the questionnaire and the application of skip logic to a question series. In the WHO 2016 tool, first-order questions are questions required of all respondents, and second-order (and subsequent order) questions are delivered dependent on the response to the first order question. For example, a "Yes" response to a first-order question might then trigger second- and third-order questions exploring this "Yes" response in further detail, whereas those who answered "No" to this first order question, would not be subject to the second-order questions. Question series that might be overly lengthy or capture redundant information might benefit from application of a skip logic or reordering of questions in order to shorten the instrument and improve acceptability of the interview by respondents as well as accuracy of the responses.

From the 14 issues, one question series identified as potentially unnecessarily lengthy is that addressing injuries. The injury series starts with the first order question, "Did (s)he suffer any injury or accident that led to his or her death?" If "Yes", this question is followed by a series of second- and third-order questions investigating the nature of the injury or accident. Regardless of whether the respondent indicated the presence of an injury, they will also be asked the remainder of the required questions in the questionnaire ascertaining other non-injury related signs or symptoms (such as cough, headache or vomiting). This may lead to unnecessarily lengthy interviews if the deceased clearly died of an injury with no other signs or symptoms. Some reasons, however, to ask subsequent questions after indication of death by injury include

to determine if the death was maternal-related or to determine if the injury was caused by an underlying medical condition.

From the primary dataset, 10% (n = 1,919) of respondents reported an injury. From the reference dataset, 85% (n = 582) of those reporting an injury were determined to have an injury as the underlying cause of death (UCoD). The median number of affirmative responses to non-injury related symptom questions was 0.8 (IQR: 0–1) among those with an injury UCoD compared to 3 (IQR: 1–4) among those without an injury UCoD. The fewer number of affirmative responses to non-injury symptom questions among those assigned an injury cause of death suggests the questionnaire might benefit from shortening or application of skip logic if the respondent indicates the presence of an injury rather than be required to complete the full questionnaire.

## Redundancy

Within the WHO questionnaire, there are questions that may ascertain identical or redundant information. Evaluation of the series of questions related to seven of the 14 issues demonstrated similar response patterns to two or more questions within each series which captured overlapping information: tobacco use, sores, breast swelling, abdominal problem, vomiting, vaccination, and baby size. Details of the analysis for two of these issues—tobacco and vomiting—are provided below as an example. In the tobacco series, the two first-order questions "Did (s)he use tobacco" and "Did (s)he smoke tobacco?" ask similar information and demonstrated >95% consistency in "Yes" and "No" responses to the two questions among respondents (S1 File). In the cognitive interviewing sample, all of those who reportedly used tobacco, smoked it. There was no one who used tobacco in some other way that was not smoking. However, answers varied for those who had quit smoking. Some answered "Yes" to the two first-order tobacco questions, and others answered "No". The qualitative cognitive interviewing results also suggested that the similarity of questions in the tobacco series was confusing or seemed repetitive to respondents; as noted by one respondent in response to the questions of whether they smoked tobacco and what kind of tobacco was used: "He used to smoke cigarettes as I said", the respondent answered.

The question series addressing the symptom vomiting begins with two first-order, required questions "Did (s)he vomit?" and "To clarify, did (s)he vomit in the week preceding death?", followed by the second order question "How long before death did (s)he vomit?". Given the two first order questions are ascertaining similar information, one of these first-order questions could potentially be eliminated. Further, two of the three questions address the timing of the vomiting, suggesting an element of redundancy. The frequencies of affirmative and negative responses to the two first-order questions were similar: 97% of respondents who reported the deceased vomited in the week before death also answered "Yes" to the first question, "Did (s)he vomit?"; 98% of respondents who answered "No" to the first question also answered "No" to the second question (Table 2) suggesting good capture of the symptom of vomiting with only one of the questions. From the qualitative review, addressing the timing of the vomiting, most respondents understood the first question, "Did (s)he vomit?", as asking about whether or not the decedent vomited in the immediate period before death, which varied from the hours before death to a few weeks prior to death.

## Frame of reference

Level of familiarity and experience with the deceased may impact a question response. Questions that are outside the frame of reference of the respondent may affect the ability to provide an accurate answer. The level of familiarity of the respondent with the deceased was evaluated

**Table 2. Crosstabulation of responses to questions, "Did (S)he vomit?" and "Did (s)he vomit in the week before death?"**

| Did (s)he vomit? | Did (s)he vomit in the week before death? | | | | |
| --- | --- | --- | --- | --- | --- |
| | n | | | | |
| | Row% | | | | |
| | Col% | | | | |
| | **Yes** | **No** | **DK** | **Ref** | **Total** |
| Yes | 3,127 | 1,080 | 72 | 3 | 4,282 (38%) |
| | 73% | 25% | 2% | <1% | |
| | 97% | 14% | 32% | 50% | |
| No | 82 | 6,700 | 32 | 0 | 6,814 (60%) |
| | 1% | 98% | 1% | | |
| | 3% | 86% | 14% | | |
| DK | 2 | 56 | 122 | 1 | 181 (2%) |
| | 1% | 31% | 67% | 1% | |
| | <1% | <1% | 54% | 17% | |
| Ref | 0 | 0 | 0 | 2 | 2 (<1%) |
| | | | | 100% | |
| | | | | 33% | |
| Total | 3,211 (28%) | 7,836 (70%) | 226 (2%) | 6 (<1%) | 11,279* |

Data source: primary dataset (n = 11,279)

* Both questions were not asked by some countries/regions

in two ways: 1) measuring the association of response patterns with the questions "What is your relationship to the deceased?" and "Did you live with the deceased?" and 2) measuring the percentage of "don't know" responses for a relevant question. Concerns relating to frame of reference of the respondent were explored for four of the 14 issues–the female health related questions, lumps, tobacco, and baby size. In evaluating six of the female health related questions, close family member respondents were significantly less likely than other respondents to provide a "Don't Know" or "Refused" response for two of the six questions, "When she had her period, did she have vaginal bleeding in between menstrual periods?" (PR 0.71, 95% CI 0.64–0.80) and "At the time of death was her period overdue?" (PR 0.74, 95%CI 0.61–0.90) (Table 3). Further, a respondent who lived with the deceased was also less likely than a respondent who did not live with the deceased to provide a "Don't Know" or "Refused" response for five of the six questions. In the series of questions related to lumps, respondents who were close family members or lived with the deceased were less likely to provide a "Don't Know" or "Refused" response to the first-order question about the presence of lumps (Close Family PR 0.51, 95%CI 0.42–0.63; Lived with Deceased PR: 0.32, 95%CI 0.26–0.41), while living with the deceased demonstrated a lower PR of non-substantive response to the question about the presence of breast swelling (PR 0.51, 95%CI 0.32–0.74) (S1 File).

Similarly, the level of detail sought in a question, such as frequency or weight, may be unknown or outside the frame of reference for a respondent. For example, 63% (n = 1,516) of respondents who reported the deceased smoked were not able to state the number of cigarettes the deceased smoked per day. Forty-one percent (n = 1,187) of responses to the birth weight question were unknown or implausible (< 100 or > 6,000 grams; many weights were recorded as < 100 grams, suggesting they were possibly recorded in units of kilograms instead of the requested units of grams). (S1 File).

**Table 3. Prevalence Ratios (PR) of "Don't Know" or "Refused" responses among respondents who were close family members of the deceased compared with other relationship to the deceased and lived with the deceased compared with did not live with the deceased.**

| Question | Percent of Don't know or Refused Response (among all respondents) | Close Family vs Other | Lived with Deceased vs Did Not Live with the Deceased |
|---|---|---|---|
| | | PR | PR |
| | | (95% CI, p value) | (95% CI, p value) |
| Did she ever have a period or menstruate? | 3% | 0.81 | 0.48 |
| | | (0.6–1.1, p = 0.13) | (0.35–0.65, p<0.001) |
| When she had her period, did she have vaginal bleeding in between menstrual periods? | 26% | 0.71 | 0.77 |
| | | (0.64–0.80, p<0.001) | (0.67–0.89, p<0.001) |
| Was the bleeding excessive? | 12% | 1.31 | 0.56 |
| | | (0.7–2.4, p = 0.33) | (0.28–1.11, p = 0.1) |
| Was there excessive vaginal bleeding in the week prior to death? | 6% | 1.23 | 0.54 |
| | | (0.9–1.5, p = 0.5) | (0.40–0.72, p<0.001) |
| Did her menstrual period stop naturally because of menopause or removal of uterus? | 6% | 1.12 | 0.48 |
| | | (0.9–1.4, p = 0.5) | (0.36–0.64, p<0.001) |
| At the time of death was her period overdue? | 26% | 0.74 | 0.70 |
| | | (0.61–0.9, p<0.05) | (0.58–0.84, p<0.001) |

Data source: primary dataset

## Clarity of construct

Clarity of construct refers to the respondent's ability to understand the terminology used in the question as well as the correct association of the construct intended in the question with that which was interpreted by the respondent. In the first case, quantitative analysis showed a lack of consistency in response patterns for questions seeking similar information but using different terminology. For example, 45% of respondents who reported the presence of a pit or ulcer on the foot reported the presence of sores. Nineteen percent (n = 7) of respondents who reported a birth weight >4500 grams reported "yes" to the questions "At birth, was the baby larger than usual?". Sixteen percent of respondents (n = 79) who reported a diagnosis of dementia in the decedent reported the presence of mental confusion (S1 File).

Incorrect association of intended constructs was observed among the medical diagnosis questions, such as those inquiring if the deceased had ever been diagnosed by a healthcare provider with dengue fever or a stroke. Qualitative cognitive interviewing results suggested two general patterns in which respondents evaluated the health conditions of the deceased: A) Medical diagnoses from a health professional or B) symptoms perceived to be related to the condition, as shown in Fig 1.

For example, some respondents confused or conflated the disease under question with another condition such as confusing dengue fever with other diseases such as yellow fever, malaria, and sickle cell anemia (pattern A2 from Fig 1). Other respondents based their answer on whether or not the decedent displayed any symptoms that they understood to be related to the condition, such as breathlessness with a diagnosis of COPD (pattern B in Fig 1). In some cases, the respondents did not know the condition or the symptoms, but still gave a "Yes" or "No" response.

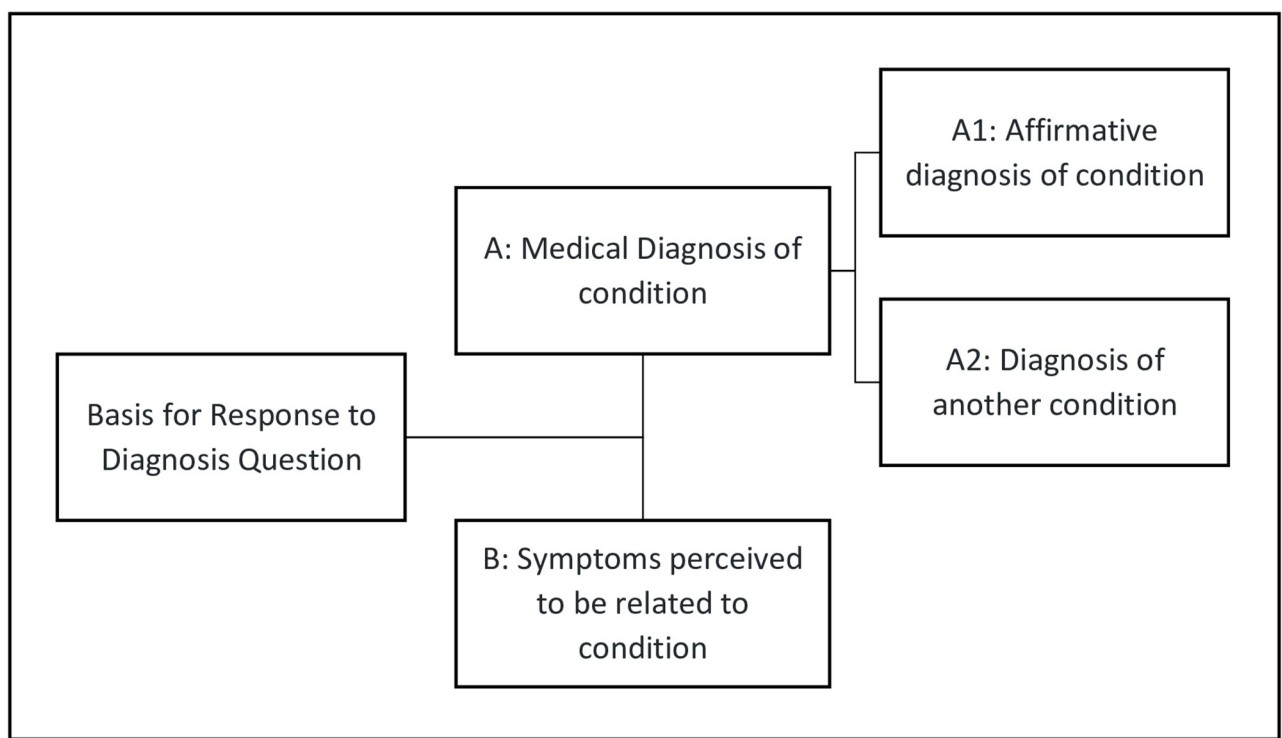

**Fig 1.** Respondent cognitive pattern for health condition questions.

## Discussion

In this investigation, we present the use of a mixed methods analysis in the evaluation of a standardized VA questionnaire. We present the results of both quantitative and qualitative analysis that can be used to inform decisions around the revision of the instrument to improve its overall accuracy and utility. Questionnaire revision decisions cannot be made from this evidence alone, as many other factors need to be taken into consideration when formally changing an instrument. From the application of the mixed methods methodology, however, we have described four broad themes that may contribute to the underperformance of 14 identified issues within the instrument: question sequence, redundancy, frame of reference and clarity of construct.

Overly lengthy questionnaires can lead to survey fatigue and decrease the acceptance of the tool among respondents [27, 28]. Application of skip or branching logic can provide opportunities to shorten a questionnaire dependent on the responses. In the injury series, the median number of affirmative responses to non-injury related symptom question was three times lower among those assigned an injury cause of death compared to those who were not. These findings suggest that deaths with indication of injury most often have no other symptoms to report and eliminating the rest of the questionnaire or using an abbreviated set of symptom questions to screen for non-injury or maternal causes after report of an injury could be applied.

The inclusion of questions seeking redundant or overlapping information can impact the answers provided by the respondent. In general, when respondents are confused about similarity among questions, such as in the tobacco section, they attach meaning to this phenomenon. Possible meanings include "I must have misunderstood one of the questions" and "they are trying to trick me", which can lead to response errors. When considering eliminating a

question in a series, the level of agreement between questions can yield useful information. Similar response patterns among questions ascertaining overlapping information suggest the feasibility of eliminating a question such as in the vomiting series. Likewise, questions with similar but distinct terminology can results in incorrect association of intended constructs, as was observed among the medical diagnosis questions. Clarity of the construct of interest can be supported by strategic organization of questions to provide a questionnaire flow that aids the respondent in tracking the distinct constructs of interest. More routinely, the open narrative that is typically also collected during the VA, where the respondent explains in their own words the circumstances of death, can also be used to verify information reported on related constructs in the "closed" section of the questionnaire. Open narratives have been reported as a way to build rapport with respondents, improving the ability to collect quality information [29]. However, further work is needed to understand how best to optimize the use of the open narrative in the VA questionnaire and in the cause of death assignment and quality control processes.

Choosing the right respondent is key in the application of a VA [1]. Level of familiarity with the deceased can affect the accuracy of the responses provided by the respondent. In our investigation, gendered questions, such as the female reproductive health questions, yielded more substantive, "Yes" or "No", responses when provided by respondents who were either close family members or lived with the deceased—the two respondent characteristics that were evaluated in this work. Understanding this difference in responses dependent on respondent characteristics is key when addressing culturally sensitive or gender-specific topics such as deaths in women of reproductive age. The most recent versions of the WHO VA questionnaire (v1.5.2 and v1.5.3) contain a question on the sex of the respondent which may provide additional information about the impact of respondent characteristics on the usefulness of responses [1]. Though not explored in this work, the relationship of the interviewer to the respondent and the community in which they are conducting VA interviews is also a known factor impacting VA performance [30].

Cognitive interviewing provided valuable insight into disparities observed in the quantitative analysis. Identification of unanticipated response patterns through the quantitative analysis highlighted the need for a better understanding of respondent knowledge and interpretation of the question construct. A review of cognitive interviewing results for questions capturing similar information, such as birth weight and medical diagnoses, but having differing response patterns, has elucidated patterns of interpretation of the construct that may lead to response errors of false positive responses. This information could be used in the questionnaire revision process to rephrase constructs within a question in order to improve accuracy.

Our investigation was subject to several limitations. With the exception of some CHAMPS data from Bangladesh, the available VA data only represent sub-Saharan Africa. Considering the variation in epidemiologic patterns and cultural practices across regions—for example types of tobacco used most frequently—the response patterns analyzed in this investigation may not fully represent those that would be expected in other regions. Also, there is variation in the way the final VA instrument is applied in given settings, due to different versions of the 2016 questionnaire being used, or other modifications that teams make, which may impact the electronic skip pattern and response frequencies. When using these quantitative and qualitative findings to revise an instrument, it is imperative to consider question requirements as well as impact of instrument changes on the question weightings used by the various automated algorithms. Alterations in the questions, question series or question elimination can affect the input into the decision matrix and the determination of the underlying cause of death from these algorithms. Furthermore, it should be noted that this investigation did not explore the

specific impact of linguistic differences, translation, or other culturally appropriate adaptations that are often made during VA implementation. While the open discussion between the respondent and the cognitive interviewer aims to capture an understanding of the respondent's cognitive processes compared to the intent of the questions, further work is needed to fully understand variations across culture and language. Finally, the sample size did not permit analysis of differences in results based on country or age group of the deaths; additional data would facilitate deeper analysis of such comparisons.

While this work demonstrates how mixed methods analysis can be used to improve VA processes, it also highlights the many additional areas in which VA methods—including the questionnaire, the interview process, and the assignment of cause of death—can continue to be improved. In addition to further work to optimize the use of the open narrative and to understand the impact of cultural and linguistic variations on VA, VA methods can also be advanced through the ongoing collection of geographically and epidemiologically representative reference deaths, against which VA data can be evaluated and knowledge of the symptom-cause relationship can be improved.

Findings from this investigation provide supporting evidence for the revision of the 2016 WHO VA instrument; specific recommendations and considerations based on the complete analysis are included in the Supplemental file (S1 File). Quantitative and qualitative analysis results can identify series of questions which could be shortened through elimination of redundancy, series of questions requiring clarification due to unclear constructs, and the impact of respondent characteristics on quality of responses. These changes can lead to better understanding of the question constructs by the respondents, increase in acceptance of the tool, and improvement in overall accuracy of the VA instrument. These findings also support the need for the selection of an appropriate respondent to the questionnaire in order to maximize accuracy of responses particularly in culturally sensitive topics and diagnoses. The integration of quantitative and qualitative data sources used in this mixed methods approach have identified areas of underperformance of the questionnaire and provided evidence to inform improvement efforts. This application has shown the mixed methods approach to be a useful methodology which can be applied in the evaluation of multiple platforms including questionnaires, surveys, and other information-gathering tools.

## Supporting information

**S1 File. Mixed-methods analysis of select issues reported in the 2016 WHO VA questionnaire: Summary report.**
(DOCX)

## Author Contributions

**Data curation:** Agbessi Amouzou, Dianna M. Blau, Debbie Bradshaw, El Marnissi Abdelilah, Pamela Groenewald, Brian Munkombwe, F. Sam Notzon, Steve Biko Odhiambo, Paul Scanlon.

**Formal analysis:** Brent Vickers, Clarissa Surek-Clark, Jordana Leitao, Dianna M. Blau, Debbie Bradshaw, El Marnissi Abdelilah, Pamela Groenewald, Chomba Mwango, Paul Scanlon.

**Investigation:** Kristen Pettrone, Brent Vickers, Hermon Gebrehiwet, Clarissa Surek-Clark, Chomba Mwango.

**Methodology:** Brent Vickers, Clarissa Surek-Clark, Jordana Leitao, Debbie Bradshaw, Pamela Groenewald, Chomba Mwango, Paul Scanlon.

**Project administration:** Erin Nichols.

**Supervision:** Erin Nichols.

**Validation:** Erin Nichols.

**Writing – original draft:** Kristen Pettrone.

**Writing – review & editing:** Erin Nichols, Brent Vickers, Hermon Gebrehiwet, Clarissa Surek-Clark, Jordana Leitao, Agbessi Amouzou, Dianna M. Blau, Debbie Bradshaw, El Marnissi Abdelilah, Pamela Groenewald, Chomba Mwango, Steve Biko Odhiambo, Paul Scanlon.

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
