## [Decision Letter · Decision Letter 0]

17 Jun 2021

PONE-D-21-12120

Mixed-Methods Analysis of Selected Issues Reported in the 2016 World Health Organization Verbal Autopsy Questionnaire

PLOS ONE

Dear Dr. Pettrone,

Thank you for submitting your manuscript to PLOS ONE. After careful consideration, we feel that it has merit but does not fully meet PLOS ONE’s publication criteria as it currently stands. Therefore, we invite you to submit a revised version of the manuscript that addresses the points raised during the review process.

We look forward to receiving your revised manuscript.

Kind regards,

Prof. Ritesh G. Menezes, M.B.B.S., M.D., Diplomate N.B.

Academic Editor

PLOS ONE

Journal Requirements:

Reviewers' comments:

Reviewer's Responses to Questions

**Comments to the Author**

1. Is the manuscript technically sound, and do the data support the conclusions?

Reviewer #1: Yes

Reviewer #2: Partly

Reviewer #3: Yes

2. Has the statistical analysis been performed appropriately and rigorously? 

Reviewer #1: Yes

Reviewer #2: Yes

Reviewer #3: I Don't Know

3. Have the authors made all data underlying the findings in their manuscript fully available?

Reviewer #1: No

Reviewer #2: No

Reviewer #3: No

4. Is the manuscript presented in an intelligible fashion and written in standard English?

Reviewer #1: Yes

Reviewer #2: Yes

Reviewer #3: Yes

5. Review Comments to the Author

Reviewer #1: This is an interesting approach to trying to determine target areas for refining the WHO's 2016 VA tool, and the use of mixed-methods is an appropriate design. Overall, some more detail in the methods, especially around the qualitative data is needed, and there are some areas that would benefit from more discussion (language, and the interviewer perspective).

Abstract:

- Can the setting be included in the methods? And a clarification about the age range for the VAs in question.

- For someone who will only read the abstract, I don't think it will be clear what you mean by item series and constructs. Appreciate the space is limited, but adding a very brief definition of these would really help readability here.

Background:

- "Feedback from end users of the WHO 2016 VA instrument has been compiled" - this needs a bit more information, as it wasn't totally clear to me whether this had been done before, who the end users were, were they from different settings to the current study etc. Given this is the entry point to the study, more clarity is needed.

- Given the comment above, the background on mixed-methods could be greatly reduced (as this is a common methodology), to keep this section concise. E.g. just keeping the content about mixed-methods for survey development.

Methods:

- I just wanted to clarify - so all the VA data was collated, and then cause of death was assigned independently by clinicians from South Africa? Did none of the data sources already have causes of death assigned, and if so, what did you do about conflicts? This process was not totally clear.

- Can you explain how (and why)the sub-sample was generate for cause of death - was it random? Why was it not stratified by age-group?

- Can you explain why Zambia and Morocco were chosen?

- What country is CCQDER based in? 'National' is not so informative for international readers!

- For the cognitive interviews, can you say something about who the 'local researchers' are (and consider re-phrasing this term), and languages e.g. were notes taken in English? Or other languages and translated? How was nuance of language/translations/understanding dealt with, given the cognitive interviewing methodology.

- "The qualitative cognitive interviewing results were analyzed using typical qualitative analysis methods" - can you be more explicit and state the approach used? Also, who conducted the analysis, and how did you think about validity?

- For triangulation, from reading it sounds like the analysis was done iteratively, so triangulation was done during the analysis phase, and not in the interpretation phase - it would be good to state this clearly (i.e. a concurrent iterative triangulation mixed-methods approach?).

Results:

- Figure 1 appeared to be distorted or missing information.

Discussion:

- When considering duplication of information (or redundancy of questions), there have been previous reports from fieldworkers that they found open narratives useful as a way to cross-check information then given to closed questions. It would be worth discussing this wider point, about recall and reliability of data, and whether having some 'checks' in the questionnaire would always be negative. I.e. could these actually be turned into quality control flags?

- Its also worth raising in the point about who the respondent is, to consider who the interviewer is, and how this also plays a role in data quality. There is some literature on this around insiders versus outsiders conducting VAs.

- On the point that nearly all data was from the African region, the discussion on tobacco could include a clarification, given smokeless tobacco use is most common in the Asian region (https://bmcmedicine.biomedcentral.com/articles/10.1186/s12916-020-01677-9).

Reviewer #2: The authors present important evaluation of the WHO VSA identifying areas where improvement can be made in this widely used tool. Given the importance of identifying causes of death as countries work to reduce preventable mortality on the road to achieving effective UHC and meeting the goals in the SGDs. The manuscript identifies 4 areas where change is needed based on unsolicited input from users (unable to assess that data source as the link is broken) and analysis of results from 2 data sets of quantitative results. The authors however then describe a mixed methods, which is a weak part of an otherwise strong paper. The description of the analysis (“typical qualitative analysis methods” which is insufficient and the references are about cognitive interviewing (one on analysis)) and requires more detail (see COREQ for describing qualitative analysis). In addition it is difficult to identify where the results were used beyond the analysis of the already submitted input. It would also be important to understand the process through which the countries undertook linguistic and cultural translation and if there were differences in the results based on country. More specific details are below. Also-data are plural and should be corrected.

Introduction:

Why is the mixed methods approach “novel”? This is hard to interpret given the paucity of description

More details on how the feedback was obtained (line 117). The link to reference 19 is broken

Methods: see above. A description of how the 2 different VSA datasets were used as this does not emerge in results. In Table 1-was this only from the submitted comments or also done through whether the new qualitative data or from analysis of the VSA results to identify additional areas beyond the 14?

Results:

Overall well described but some language is hard to follow (such as the discussion of the vomiting questions-what was the conclusion-which to drop? It would also be helpful for others such as respondent type to know if and how much value of still asking (the PR shows relative value but what % of non-close relatives for example, did answer yes or no”. Similarly (from discussion) -is there a potential difference behind accuracy (correct answer) and assurance (willing to say yes/no even if they do not know due to sense they should know) The information for example on responses to birth weight-what is ‘too low” a definitive response rate to warrant dropping for that respondent type. For the clarity of construct-where there were differing responses, is there anything from the second VSA data set to identify which questions was closer to reality?. Figure 1 is also a bit hard t understand

Discussion line 374-377: It was unclear what cognitive evaluation was done and what patterns could results in false (+)

In the limitations, the comments about mixed methods is hard to interpret given the issues around description of methods and results noted above. Also there is no discussion at differences between processes for linguistic and cultural translation as well as the cultural adaptations needed.

Reviewer #3: Mixed-Methods Analysis of Selected Issues Reported in the 2016 World Health Organization Verbal Autopsy Questionnaire

PONE-D-21-12120

OVERALL COMMENTS

Thank you for the opportunity to review this important paper advancing the international VA standard interview towards a format that is more amenable to widespread application. The paper reports an important and original contribution and is worthy of publication. There are several overall and specific comments below, all relatively minor but may strengthen the paper in terms of clarity and meaning to readers unfamiliar with VA.

Overall, I found several key pieces of information to be slightly opaque. In the description of the ’14 issues’ it would be useful to summarise what these issues were – i.e., that they relate to process (generally shortening the interview) and substantive issues (confusion with constructs/meaningful responses). The selection/origins of the data and sequence of the analysis were also not entirely clear e.g., whether the 14 issues from VA end-users is part of or separate to this analysis. These points could be introduced and explained with more clarity, and earlier in the paper. Moreover, some attention to where the ’14 issues’ came from; what a cognitive interview is; what the qualitative work did (content of the interview); and, in the abstract, an overview of the substantive findings.

A statement on objectives was also missing from the abstract and paper. This could be e.g., related to identifying: ‘underperformance or redundancy [of items] within the instrument’ (page 17, line 204) and ideally connected to higher order aims ‘to shorten the instrument and improve acceptability of the interview by respondents as well as the accuracy of the responses’ (page 18, lines 218-9) or ‘to inform decisions around the revision of the instrument to improve its overall accuracy and utility’ (page 24, lines 338-9).

Consistency with key terms would also be useful. There was some variety in terms used especially for the qualitative elements (e.g., cognitive testing results and typical qualitative analysis methods) consistency with these would be preferable.

I hope these comments are of use in revising and clarifying some key aspects. Finally, it should be noted that I am not a quantitative methodologist and a reviewer with skills in this area should also review the paper.

SPECIFIC COMMENTS

ABSTRACT

1. Page 10, line 56-7: It would be useful to summarise what the issues are, and provide a stated objective.

2. Page 10, line 59: It would be useful to indicate where the VAs were drawn from.

3. Page 10, line 60: How is the quality of responses defined?

4. Page 10, line 61: For the unfamiliar reader, it would be useful to understand what a ‘cognitive interview’ is.

5. Page 10, line 62: It would be useful to understand, in the abstract, the settings from which the data were drawn.

6. Page 10, line 63: It would be useful to understand how identification of common themes relates to the overall aims and objectives.

7. Page 10, line 67: As per comment no. 1, it would be useful to have a summary of what the previously identified issues are. The sentence ‘Two of the 14 question series identified issues related to item sequence; seven demonstrated similar response patterns among questions within each series capturing overlapping information’ suggests what the issues relate to, but is not entirely clear.

8. Page 10, lines 69-70: Is it possible to report the respondent characteristics? Similarly on constructs outside the frame of reference?

9. Page 11, lines 74-8: As per comments above, the substantive results feel lacking. Some of this content could be used to develop aims and objectives.

INTRODUCTION

10. Page 12, line 99: It might be useful, for the unfamiliar reader, to understand what is meant by ‘item and unit’.

11. Page 12, line 105: Again, for the unfamiliar reader, it might be useful to briefly summarise how different philosophical positions on truth and knowledge underpin different methodologies. Does this only improve reliability?

12. Page 12, line 112: As above, please include a brief description of ‘cognitive interview’.

13. Page 12, line 117: Who are the ‘end-users’?

14. Page 12, line 118: As per comments on abstract, please provide a summary of the 14 issues, from whose perspectives and using what approaches these issues were identified.

15. Strongly suggest that the authors articulate aims and objectives.

METHODS

Quantitative data collection

16. Page 13, line 129: (And throughout) consistency with abbreviations needed.

17. Page 13, paragraph 2: Great to understand the settings from which data derived. It would be useful to understand how and why data from these countries were included.

18. Page 14, lines 148-53: A short explanation of the relationship between the primary and reference datasets would be useful.

Qualitative data collection

19. Page 14, line 156: As above – a) why these settings? And b) what is a cognitive interview?

20. A description of what the cognitive interview sought information on would be useful to include. The authors may also wish to report on key information such as: How long did the interview take? Was it structured/semi-structured? Is the interview guide available? How many interviews were done in each setting?

Analysis

21. Page 15, line 181: As above, who are end users?

22. Page 15, lines 184-5: Great to have the 14 problematic areas, a description of this could come earlier, however. It is also not clear whether the 14 issues from VA end-users is part of or separate to this analysis.

23. This section might usefully be revised to state specifically the aspects being assessed, how these relate to the ’14 issues’ and how the assessment allowed the issue to be addressed.

24. Page 16, line 193: what does ‘typical qualitative analysis methods’ mean? Details on the specifics of the analytical approach, and why the approach was appropriate would be useful to include.

25. Page 16, lines 194-5: ‘cognitive interviewing data’ – does this mean qualitative data? It is not clear why end-users (presumably administrators of VA) would report the same or similar issues to VA respondents. Specifics of the quantitative analysis performed on the inductive analysis would be useful to include.

26. Page 16, line 197: ‘underperformance of the item’ gives some sense of the overall objective and how the analysis contributed to achieving, however this could be brought out more clearly.

27. Table 1 – please number the 14 items. In the description, it might be useful to summarise that these relate to process (repetition, response patterns, or shortening of the interview) and substantive issues (confusion with constructs/consistent and meaningful responses). As above, this could be introduced and explained with more clarity, and earlier in the paper.

RESULTS

28. Page 18, line 203: Consistency with ‘concepts’ and ‘constructs’ in reference to items in the interview would be useful. Considering much of the analysis refers to respondents’ understanding of constructs, the authors may wish to refer to ‘four broad themes’, here.

29. Page 18, lines 204-5: The authors may wish to indicate that ‘overlap within the item series’ is understood as ‘redundancy’.

Redundancy

30. Page 19, line 243: some explanation of ‘seven of the question series’ would be useful to include.

31. This section opens with a statement about question series on tobacco, sores, breast swelling, abdominal problem, vomiting, vaccination, and birth weight. It is not clear why results of the analysis of response patterns are presented in detail for one of these (vomiting) in an appendix for another (tobacco use), triangulating with the qualitative analysis for one (tobacco use) and not for the others.

32. Page 21, line 272: does ‘cognitive testing results’ mean qualitative analysis? Various terms are used for this element of the analysis, which may not be entirely clear to readers.

Frame of reference

33. Page 21, line 289: Again, ‘question series’ would be useful to describe to the unfamiliar reader.

34. Table 3: it would be useful to understand why PRs are presented for 6 questions. What about the others?

Clarity of construct

35. Page 23, lines 312-4: The difference between the two elements of clarity of construct is unclear.

36. Page 23, lines 314-5: The sentence ‘In the qualitative analysis, items seeking similar information but using different terminology, or items having overlapping constructs demonstrated differing response patterns’ is slightly unclear, suggest revise in the active voice.

37. Page 23, line 323: Again, the term ‘cognitive testing results’ is used. This term is only introduced in the results section. It is perfectly acceptable to use the term, however it should be introduced and described in the methods section and used consistently thereafter.

38. As above, the triangulation and choice of specific results presented is unclear.

DISCUSSION

39. Page 24, line 341: See point above, the authors may wish to consistently refer to themes from the mixed methods analysis. Various reference to constructs and concepts may be confusing for readers.

40. Page 25, paragraphs 1-2: As above, were these the findings of note from the item sequence analyses? ‘Such as’ indicates there were others.

41. Page 25, paragraph 3: Was there any attempt to examine response patterns by respondent type? Or by setting?

42. Page 26, lines 371-4: As above, consistency with key terms – ‘cognitive testing’, used for the first time in the results and frequently thereafter, and here for the first time, ‘cognitive evaluation’, and prior with qualitative analysis could be confusing for readers unfamiliar with these methods. Introducing and explaining key terms in the methods section, and carrying these through the paper consistently would further strengthen the reporting of the research process and findings.

43. The discussion could include some attention to the wider debates on VA. How does, for example, this research contribute to the methodological transition of the method?

44. Page 26, paragraph 2: The limitations are useful and relate to some comments above on how study settings were selected, and where data were drawn from, which, if raised in the methods, could be critically reflected on here. The authors may also wish to consider strengths of the approach, and future directions. Also, on page 27 (line 397) the approach is described as novel. It would be useful to understand what type of research or other information has informed previous iterations of the instrument, and how this approach is new/contributes to what has gone before.

45. Page 26, paragraph 2: While it is more customary in qualitative research, the authors may wish to reflect on their positionality and how this influenced the research process and results.

46. Pages 26-7, lines 389-99: This reads as a useful conclusion. Is this section required for this type of paper, in this journal?

47. Pages 26-7, lines 389-99: Does the statement ‘Questionnaire revision decisions cannot be made from this evidence alone’ (page 24, line 339) contradict the subsequent statement ‘Findings from this investigation provide supporting evidence for the revision of the 2016 WHO verbal autopsy instrument.’ (page 26, lines 389-90)? See above, the authors may wish to consider articulating a series of directions for future research to inform decisions on revision of this instrument.

END OF REVIEW

6. PLOS authors have the option to publish the peer review history of their article (what does this mean?). If published, this will include your full peer review and any attached files.

Reviewer #1: No

Reviewer #2: No

Reviewer #3: **Yes: **Lucia D'Ambruoso

---

## [Author Response · Author response to Decision Letter 0]

5 Jan 2022

Response to Reviewers

Journal Requirements:

DONE

Regarding the availability of the data, indeed, we are only able to consider data sharing by request. While the data are de-identified, some of the data are sourced from sensitive data that contribute to vital statistics. The data have been provided by teams for this analysis with the agreement that the data will only be used by the agreed collaborators for the purpose of contributing to the improvement of the WHO verbal autopsy questionnaire. Data contributors agreed to the publication of aggregate findings and are co-authors of the paper. We can address any requests for the data individually; requests can be directed to myself (corresponding author), and we will work with co-authors/data contributors as needed to seek the appropriate permissions if and as need. 

DONE

Reviewers' comments:

Reviewer's Responses to Questions

Comments to the Author – responses provided in italics 

5. Review Comments to the Author

Reviewer #1: This is an interesting approach to trying to determine target areas for refining the WHO's 2016 VA tool, and the use of mixed-methods is an appropriate design. Overall, some more detail in the methods, especially around the qualitative data is needed, and there are some areas that would benefit from more discussion (language, and the interviewer perspective).

Abstract:

- Can the setting be included in the methods? And a clarification about the age range for the VAs in question. – details have been added

- For someone who will only read the abstract, I don't think it will be clear what you mean by item series and constructs. Appreciate the space is limited, but adding a very brief definition of these would really help readability here.- have reworked how these points are made to be more clear (throughout)

Background:

- "Feedback from end users of the WHO 2016 VA instrument has been compiled" - this needs a bit more information, as it wasn't totally clear to me whether this had been done before, who the end users were, were they from different settings to the current study etc. Given this is the entry point to the study, more clarity is needed. – Have added clarification that end users are any field teams that use the VA questionnaire; have also added detail about the platform used to compile feedback. 

- Given the comment above, the background on mixed-methods could be greatly reduced (as this is a common methodology), to keep this section concise. E.g. just keeping the content about mixed-methods for survey development. – the background information included briefly describes the mixed-methods approaches that underpin this work, and thus we propose to keep it; per another reviewer’s suggestion, we have added additional detail on qualitative and quantitative approaches. 

Methods:

- I just wanted to clarify - so all the VA data was collated, and then cause of death was assigned independently by clinicians from South Africa? Did none of the data sources already have causes of death assigned, and if so, what did you do about conflicts? This process was not totally clear. -No; VA questionnaire data was available from 5 project teams; additional information on cause of death, as assigned by a physician, was available from 1 project team (South Africa). Clarification added in the first sentence of the quantitative data collection section.

- Can you explain how (and why) the sub-sample was generate for cause of death - was it random? Why was it not stratified by age-group? – per the above, it was not a random sample; it was based on what data was available from the project teams. 

- Can you explain why Zambia and Morocco were chosen? – Clarification added in the second sentence of the qualitative data collection section (“These sites were selected based on their readiness of the sites, given that the VA process had been well established, and the field teams were interested into evaluating the performance of the interview process using cognitive interviewing.”) 

- What country is CCQDER based in? 'National' is not so informative for international readers! – the U.S.; clarification added.

- For the cognitive interviews, can you say something about who the 'local researchers' are (and consider re-phrasing this term), and languages e.g. were notes taken in English? Or other languages and translated? How was nuance of language/translations/understanding dealt with, given the cognitive interviewing methodology. Clarification has been added through the section to clarify who the researchers were (interviewers selected from the local communities) and the role of language and translation throughout the process. The cognitive interviewers were depended on to translate local language of the respondents into English for analysis.

- "The qualitative cognitive interviewing results were analyzed using typical qualitative analysis methods" - can you be more explicit and state the approach used? Also, who conducted the analysis, and how did you think about validity --Detail has been added in the qualitative section about the iterative 5-step synthesis and reduction process, within which results are validated through comparison and iteration of findings; results were further validated in the mixed-methods triangulation. For triangulation, from reading it sounds like the analysis was done iteratively, so triangulation was done during the analysis phase, and not in the interpretation phase - it would be good to state this clearly (i.e. a concurrent iterative triangulation mixed-methods approach?). –Correct, triangulation was iterative, and clarification has been added.

Results:

- Figure 1 appeared to be distorted or missing information. – redid figure to clarify relationship of content.

Discussion:

- When considering duplication of information (or redundancy of questions), there have been previous reports from fieldworkers that they found open narratives useful as a way to cross-check information then given to closed questions. It would be worth discussing this wider point, about recall and reliability of data, and whether having some 'checks' in the questionnaire would always be negative. I.e. could these actually be turned into quality control flags? – Agree. Have added a statement about the value and potential use of the narrative in this capacity, also noting that more work is needed to understand how to make the best use of the narrative.

- Its also worth raising in the point about who the respondent is, to consider who the interviewer is, and how this also plays a role in data quality. There is some literature on this around insiders versus outsiders conducting VAs. – noted. This point with a relevant reference has been added to the section on choosing the right respondent 

- On the point that nearly all data was from the African region, the discussion on tobacco could include a clarification, given smokeless tobacco use is most common in the Asian region (https://bmcmedicine.biomedcentral.com/articles/10.1186/s12916-020-01677-9). – noted; this example has been added in the limitations section with a broader explanation about the impact of the variation in epidemiologic patterns and cultural practices across regions.

Reviewer #2: The authors present important evaluation of the WHO VSA identifying areas where improvement can be made in this widely used tool. Given the importance of identifying causes of death as countries work to reduce preventable mortality on the road to achieving effective UHC and meeting the goals in the SGDs. The manuscript identifies 4 areas where change is needed based on unsolicited input from users (unable to assess that data source as the link is broken) and analysis of results from 2 data sets of quantitative results. The authors however then describe a mixed methods, which is a weak part of an otherwise strong paper. The description of the analysis (“typical qualitative analysis methods” which is insufficient and the references are about cognitive interviewing (one on analysis)) and requires more detail (see COREQ for describing qualitative analysis). In addition it is difficult to identify where the results were used beyond the analysis of the already submitted input. It would also be important to understand the process through which the countries undertook linguistic and cultural translation and if there were differences in the results based on country. More specific details are below. Also-data are plural and should be corrected. – have addressed these comments as described below.

Introduction:

Why is the mixed methods approach “novel”? This is hard to interpret given the paucity of description

More details on how the feedback was obtained (line 117). ¬-The application of the mixed methods approach to improving the WHO VA questionnaire is novel, but for simplicity, this term has been removed. 

The link to reference 19 is broken – have updated: https://github.com/SwissTPH/WHO_VA_2016

Methods: see above. A description of how the 2 different VSA datasets were used as this does not emerge in results. –Clarification has been added in the quantitative section regarding the composition of the primary and reference datasets. In Table 1-was this only from the submitted comments or also done through whether the new qualitative data or from analysis of the VSA results to identify additional areas beyond the 14? – only from the submitted comments. 

Results:

Overall well described but some language is hard to follow (such as the discussion of the vomiting questions-what was the conclusion-which to drop? – Addressing comments from other reviewers, have added modifications throughout to clarify the language. However, to note, for simplicity, we are not providing specific recommendations with the examples here—they are in the supplemental file, and we’ve added a sentence on this in the last paragraph of the discussion. It would also be helpful for others such as respondent type to know if and how much value of still asking (the PR shows relative value but what % of non-close relatives for example, did answer yes or no”. – have added the overall percent of refused and don’t know responses for each question for consideration—where there is a higher percentage of don’t know/refused, the value of the question may be questioned. (this was considered for all questions at a later phase in the broader questionnaire revision process). Similarly (from discussion) -is there a potential difference behind accuracy (correct answer) and assurance (willing to say yes/no even if they do not know due to sense they should know) – Indeed, and that is a purpose of the quantitative comparisons made in this analysis- to compare response patterns across related questions to detect deviations from what is expected if the questions were answered as intended. The information for example on responses to birth weight-what is ‘too low” a definitive response rate to warrant dropping for that respondent type. – Have added detail on the low and high cutoffs used to determine plausibility. For the clarity of construct-where there were differing responses, is there anything from the second VSA data set to identify which questions was closer to reality? – Agree this would be a relevant analysis to conduct, though it was not conducted at this phase of the analysis; have added a comment in Discussion about the value of reference deaths to support further performance evaluation. Figure 1 is also a bit hard t understand – redid figure to clarify relationship of components

Discussion line 374-377: It was unclear what cognitive evaluation was done and what patterns could results in false (+) ¬– this refers to the cognitive interviewing that was the source of the qualitative data used in the analysis; the terminology has been changed for consistency and clarity; the reference to false positives refers to the issue described in figure 1, which shows a cognitive pattern of how respondents interpreted diagnosis questions in different ways, suggesting that some respondents may have conflated the presence of symptoms with a diagnosis of a related health condition, yielding a potential false positive response. In the limitations, the comments about mixed methods is hard to interpret given the issues around description of methods and results noted above. -addressed as noted above. Also there is no discussion at differences between processes for linguistic and cultural translation as well as the cultural adaptations needed. – this point has now been addressed in the limitations section

Reviewer #3: Mixed-Methods Analysis of Selected Issues Reported in the 2016 World Health Organization Verbal Autopsy Questionnaire

OVERALL COMMENTS

Thank you for the opportunity to review this important paper advancing the international VA standard interview towards a format that is more amenable to widespread application. The paper reports an important and original contribution and is worthy of publication. There are several overall and specific comments below, all relatively minor but may strengthen the paper in terms of clarity and meaning to readers unfamiliar with VA.

Overall, I found several key pieces of information to be slightly opaque. In the description of the ’14 issues’ it would be useful to summarise what these issues were – i.e., that they relate to process (generally shortening the interview) and substantive issues (confusion with constructs/meaningful responses). The selection/origins of the data and sequence of the analysis were also not entirely clear e.g., whether the 14 issues from VA end-users is part of or separate to this analysis. These points could be introduced and explained with more clarity, and earlier in the paper. Moreover, some attention to where the ’14 issues’ came from; what a cognitive interview is; what the qualitative work did (content of the interview); and, in the abstract, an overview of the substantive findings.

A statement on objectives was also missing from the abstract and paper. This could be e.g., related to identifying: ‘underperformance or redundancy [of items] within the instrument’ (page 17, line 204) and ideally connected to higher order aims ‘to shorten the instrument and improve acceptability of the interview by respondents as well as the accuracy of the responses’ (page 18, lines 218-9) or ‘to inform decisions around the revision of the instrument to improve its overall accuracy and utility’ (page 24, lines 338-9).

Consistency with key terms would also be useful. There was some variety in terms used especially for the qualitative elements (e.g., cognitive testing results and typical qualitative analysis methods) consistency with these would be preferable.

I hope these comments are of use in revising and clarifying some key aspects. Finally, it should be noted that I am not a quantitative methodologist and a reviewer with skills in this area should also review the paper. – All comments have been address as described below.

SPECIFIC COMMENTS

ABSTRACT

1. Page 10, line 56-7: It would be useful to summarise what the issues are, and provide a stated objective. – Have modified to address these issues and have named specific issues in the relevant statements in the Results section

2. Page 10, line 59: It would be useful to indicate where the VAs were drawn from. – have added

3. Page 10, line 60: How is the quality of responses defined? -rephrased to more accurately describe the association measured

4. Page 10, line 61: For the unfamiliar reader, it would be useful to understand what a ‘cognitive interview’ is- have added a brief explanation

5. Page 10, line 62: It would be useful to understand, in the abstract, the settings from which the data were drawn. – have added countries

6. Page 10, line 63: It would be useful to understand how identification of common themes relates to the overall aims and objectives. –While the specific findings for each question need to be considered for the questionnaire revision process, the four common themes are useful to summarize the findings overall and to consider how these broad themes may relate to other possible revisions (no change recommended)

7. Page 10, line 67: As per comment no. 1, it would be useful to have a summary of what the previously identified issues are. The sentence ‘Two of the 14 question series identified issues related to item sequence; seven demonstrated similar response patterns among questions within each series capturing overlapping information’ suggests what the issues relate to, but is not entirely clear. – have added names of specific issues for clarity

8. Page 10, lines 69-70: Is it possible to report the respondent characteristics? Similarly on constructs outside the frame of reference? – have added

9. Page 11, lines 74-8: As per comments above, the substantive results feel lacking. Some of this content could be used to develop aims and objectives.- have added detail as noted above

INTRODUCTION

10. Page 12, line 99: It might be useful, for the unfamiliar reader, to understand what is meant by ‘item and unit’. – agree these are confusion; have changed item to question and deleted unit

11. Page 12, line 105: Again, for the unfamiliar reader, it might be useful to briefly summarise how different philosophical positions on truth and knowledge underpin different methodologies. – Have added the requested detail after line 105. Does this only improve reliability? – No, have broadened terms used to “produce a better understanding of findings”.

12. Page 12, line 112: As above, please include a brief description of ‘cognitive interview’. – Have added at first mention of cognitive interviewing in previous paragraph. 

13. Page 12, line 117: Who are the ‘end-users’? –have added clarify that end users are any field teams that use the questionnaire

14. Page 12, line 118: As per comments on abstract, please provide a summary of the 14 issues, from whose perspectives and using what approaches these issues were identified. –additional detail has been added to clarify how the issues were identified and why they were selected for the mixed methods analysis; we have revised the background paragraph to include more detail on the platform for collecting feedback and to indicate that this paper is about the issues for which a mixed methods approach is appropriate 

15. Strongly suggest that the authors articulate aims and objectives. – Have revised the final paragraph in the background section to clarify the purpose for the investigation.

METHODS

Quantitative data collection

16. Page 13, line 129: (And throughout) consistency with abbreviations needed. -reviewed/updated

17. Page 13, paragraph 2: Great to understand the settings from which data derived. It would be useful to understand how and why data from these countries were included. – added detail (field teams that had completed at least 1,000 VAs using the 2016 WHO VA questionnaire and who agreed to contribute their data for the exercise)

18. Page 14, lines 148-53: A short explanation of the relationship between the primary and reference datasets would be useful. – have added clarification. (“The reference dataset included the VA data as in the primary dataset combined with cause of death information determined by physician review of the verbal autopsy.”)

19. Page 14, line 156: As above – a) why these settings? – added (“These sites were selected based on their readiness, given that the VA process had been well established and the field teams were interested in evaluating the performance of the interview process using cognitive interviewing.”) And b) what is a cognitive interview? – Has been added in background

20. A description of what the cognitive interview sought information on would be useful to include. The authors may also wish to report on key information such as: How long did the interview take? Was it structured/semi-structured? Is the interview guide available? How many interviews were done in each setting? – Requested details have been added in the qualitative section.

Analysis

21. Page 15, line 181: As above, who are end users? – addressed above and use deleted in this section, as all detail on feedback moved to background

22. Page 15, lines 184-5: Great to have the 14 problematic areas, a description of this could come earlier, however. It is also not clear whether the 14 issues from VA end-users is part of or separate to this analysis. – Agreed. Have moved all description of 14 issues to background.

23. This section might usefully be revised to state specifically the aspects being assessed, how these relate to the ’14 issues’ and how the assessment allowed the issue to be addressed. – Detail has been added in Analysis section to clarify the aspects assessed and relation to 14 issues. 

24. Page 16, line 193: what does ‘typical qualitative analysis methods’ mean? Details on the specifics of the analytical approach, and why the approach was appropriate would be useful to include. – Details have been added in qualitative section (iterative 5-step synthesis and reduction process).

25. Page 16, lines 194-5: ‘cognitive interviewing data’ – does this mean qualitative data? It is not clear why end-users (presumably administrators of VA) would report the same or similar issues to VA respondents. Specifics of the quantitative analysis performed on the inductive analysis would be useful to include. – this specific statement has been removed; additional detail added in the section to clarify that qualitative data refers to cognitive interviewing data; end users (which has been clarified to be field teams) are those that administer the VA process—they compile feedback from interviewers, who can report when respondents have issues understanding the questions; of course, specific cognitive testing assessments aim to systematically compile such information directly from respondents. 

26. Page 16, line 197: ‘underperformance of the item’ gives some sense of the overall objective and how the analysis contributed to achieving, however this could be brought out more clearly. – noted; this has been added in the last part of the intro section with additional information in the analysis section noting results are used to make recommendations for improvement.

27. Table 1 – please number the 14 items. In the description, it might be useful to summarise that these relate to process (repetition, response patterns, or shortening of the interview) and substantive issues (confusion with constructs/consistent and meaningful responses). As above, this could be introduced and explained with more clarity, and earlier in the paper. – numbers added in Table 1. Detail added in background where 14 issues are described about how the issues relate to the interview. Table 1 moved to after Intro.

RESULTS

28. Page 18, line 203: Consistency with ‘concepts’ and ‘constructs’ in reference to items in the interview would be useful. Considering much of the analysis refers to respondents’ understanding of constructs, the authors may wish to refer to ‘four broad themes’, here. – Noted. Changes have been made for clarity; four broad concepts/constructs changed to themes throughout; where “concepts” referred to “questions,” wording was changed to “questions.” 

29. Page 18, lines 204-5: The authors may wish to indicate that ‘overlap within the item series’ is understood as ‘redundancy’. – done, clarity added

Redundancy

30. Page 19, line 243: some explanation of ‘seven of the question series’ would be useful to include. – rephrased for clarity

31. This section opens with a statement about question series on tobacco, sores, breast swelling, abdominal problem, vomiting, vaccination, and birth weight. It is not clear why results of the analysis of response patterns are presented in detail for one of these (vomiting) in an appendix for another (tobacco use), triangulating with the qualitative analysis for one (tobacco use) and not for the others. – noted. Select examples are shown throughout the results section to demonstrate results of the mixed methods analysis contributing to the summary findings of the four broad themes contributing to issues in the questionnaire, with reference to the Supplemental file for the full analysis. This is noted at the end of the first results paragraph. Wording has been added in the redundancy section to clarify that details for two of the issues are included as an example. The two examples have been reordered to be ordered consistently with how they are mentioned in Table 1 and in the list of 7 issues with problems of redundancy.

32. Page 21, line 272: does ‘cognitive testing results’ mean qualitative analysis? Various terms are used for this element of the analysis, which may not be entirely clear to readers. – yes; have switched “testing” to “interviewing” for consistency and added “qualitative” before cognitive interviewing in this section for clarity. 

Frame of reference

33. Page 21, line 289: Again, ‘question series’ would be useful to describe to the unfamiliar reader. – noted; have made modifications for clarity in the methods/analysis section and in this section. 

34. Table 3: it would be useful to understand why PRs are presented for 6 questions. What about the others? – We used two measures to evaluate frame of reference; examples provided demonstrate each; have added a sentence clarifying these two measures in the section. 

Clarity of construct

35. Page 23, lines 312-4: The difference between the two elements of clarity of construct is unclear. – have revised to add clarity that first element refers to the ability of the respondent to understand the terminology (versus the second, which refers to the respondent responding to the intended construct). 

36. Page 23, lines 314-5: The sentence ‘In the qualitative analysis, items seeking similar information but using different terminology, or items having overlapping constructs demonstrated differing response patterns’ is slightly unclear, suggest revise in the active voice. – Have revised for clarity.

37. Page 23, line 323: Again, the term ‘cognitive testing results’ is used. This term is only introduced in the results section. It is perfectly acceptable to use the term, however it should be introduced and described in the methods section and used consistently thereafter. – modified as noted above—now using “cognitive interviewing” consistently throughout.

38. As above, the triangulation and choice of specific results presented is unclear. – Have added detail in the first paragraph of the results section to clarify that select examples are shown to demonstrate the application of different types of analysis. Have also reworked the intro statement for the first two paragraphs of this section to add clarity.

DISCUSSION

39. Page 24, line 341: See point above, the authors may wish to consistently refer to themes from the mixed methods analysis. Various reference to constructs and concepts may be confusing for readers. – Noted, have changed as recommended.

40. Page 25, paragraphs 1-2: As above, were these the findings of note from the item sequence analyses? ‘Such as’ indicates there were others. ¬¬-Paragraphs 2-5 of the Discussion provide additional context for each of the 4 broad themes, referencing examples described in the results.

41. Page 25, paragraph 3: Was there any attempt to examine response patterns by respondent type? Or by setting? ¬¬It’s unclear what is meant by respondent type. We did not examine response patterns by setting—the analysis was conducted on the full set of data; we did examine response patterns by relationship of the respondent to the deceased (as described in “frame of reference” section).

42. Page 26, lines 371-4: As above, consistency with key terms – ‘cognitive testing’, used for the first time in the results and frequently thereafter, and here for the first time, ‘cognitive evaluation’, and prior with qualitative analysis could be confusing for readers unfamiliar with these methods. Introducing and explaining key terms in the methods section, and carrying these through the paper consistently would further strengthen the reporting of the research process and findings. – use of “cognitive testing” and “cognitive evaluation” has been removed throughout. 

43. The discussion could include some attention to the wider debates on VA. How does, for example, this research contribute to the methodological transition of the method? – Have added a paragraph on this in the discussion, calling for additional work in various areas

44. Page 26, paragraph 2: The limitations are useful and relate to some comments above on how study settings were selected, and where data were drawn from, which, if raised in the methods, could be critically reflected on here. The authors may also wish to consider strengths of the approach, and future directions. Also, on page 27 (line 397) the approach is described as novel. It would be useful to understand what type of research or other information has informed previous iterations of the instrument, and how this approach is new/contributes to what has gone before. – have expanded details on limitations of where data were from in limitations; have added a paragraph (second to last in the Discussion) for future work; cognitive testing was also used to inform previous revisions of the instrument, but for simplicity, have removed the term “novel”. 

45. Page 26, paragraph 2: While it is more customary in qualitative research, the authors may wish to reflect on their positionality and how this influenced the research process and results. –this was not included with the qualitative portion of the findings from which this work was drawn; the co-authors are happy to discuss further if needed.

46. Pages 26-7, lines 389-99: This reads as a useful conclusion. Is this section required for this type of paper, in this journal? Unclear- believe discussion sections starting with a summary of results are customary?

47. Pages 26-7, lines 389-99: Does the statement ‘Questionnaire revision decisions cannot be made from this evidence alone’ (page 24, line 339) contradict the subsequent statement ‘Findings from this investigation provide supporting evidence for the revision of the 2016 WHO verbal autopsy instrument.’ (page 26, lines 389-90)? See above, the authors may wish to consider articulating a series of directions for future research to inform decisions on revision of this instrument. –These findings were one of a set of criteria that were taken into consideration during the overall questionnaire revision process (other criteria included the significance of a symptom in assigning a cause of death, as measured by empirical data and by medical expert opinion). As noted above, we have added a paragraph on future direction.

END OF REVIEW

---

## [Decision Letter · Decision Letter 1]

28 Mar 2022

PONE-D-21-12120R1

Mixed-Methods Analysis of Selected Issues Reported in the 2016 World Health Organization Verbal Autopsy Questionnaire

PLOS ONE

Dear Dr. Nichols,

Thank you for submitting your manuscript to PLOS ONE. After careful consideration, we feel that it has merit but does not fully meet PLOS ONE’s publication criteria as it currently stands. Therefore, we invite you to submit a revised version of the manuscript that addresses the points raised during the review process.

Please submit your revised manuscript by May 12, 2022. Please include the following items when submitting your revised manuscript:

A 'Response to Reviewers' letter that responds to each point raised by the academic editor and reviewer(s). You should upload this letter as a separate file labeled 'Response to Reviewers'.A marked-up copy of your manuscript that highlights changes made to the original version. You should upload this as a separate file labeled 'Revised Manuscript with Track Changes'.An unmarked version of your revised paper without tracked changes. You should upload this as a separate file labeled 'Manuscript'.

We look forward to receiving your revised manuscript.

Kind regards,

Prof. Ritesh G. Menezes, M.B.B.S., M.D., Diplomate N.B.

Academic Editor

PLOS ONE

Journal Requirements:

Reviewers' comments:

Reviewer's Responses to Questions

**Comments to the Author**

1. If the authors have adequately addressed your comments raised in a previous round of review and you feel that this manuscript is now acceptable for publication, you may indicate that here to bypass the “Comments to the Author” section, enter your conflict of interest statement in the “Confidential to Editor” section, and submit your "Accept" recommendation.

Reviewer #1: All comments have been addressed

Reviewer #2: (No Response)

Reviewer #4: (No Response)

2. Is the manuscript technically sound, and do the data support the conclusions?

Reviewer #1: Yes

Reviewer #2: Yes

Reviewer #4: Yes

3. Has the statistical analysis been performed appropriately and rigorously? 

Reviewer #1: Yes

Reviewer #2: Yes

Reviewer #4: Yes

4. Have the authors made all data underlying the findings in their manuscript fully available?

Reviewer #1: No

Reviewer #2: Yes

Reviewer #4: Yes

5. Is the manuscript presented in an intelligible fashion and written in standard English?

Reviewer #1: Yes

Reviewer #2: Yes

Reviewer #4: Yes

6. Review Comments to the Author

Reviewer #1: Thank you for the responses and the methods and origin of the different data sources are now much clearer - again, really interesting work! The most minor of things - in the abstract, the final sentence in the background should be removed, as its a conclusion statement (although I think you added this on reviewer 3's request?), so maybe an editorial decision.

Reviewer #2: The authors have done a careful job in responding to the 3 reviewers in-depth and relevant comments. This is an important manuscript which has hopefully already influenced work to continue to strengthen the VSA tool. However there remain a few targeted areas where the paper could be strengthened for reader understanding and impact

Abstract: The explanation of mixed methods is not needed, as now a commonly used and can be dropped to allow for more detail (like which 2 countries were used for the qualitative work)

In the introduction the authors provide a nice description of the uses of mixed methods, but do not then state clearly which of these is being applied

In the description of the GitHub data, they note "14 problematic issues for which solutions could be well-informed...."-I think these are 14 areas rather than grouping of issues? this should be clarified

While edited, it is still not clear to a new reader that only one of the datasets as physician validation. “The reference dataset included the VA data as in the primary dataset combined with cause of death information determined by physician review of the VA interview (PCVA or physician certified VA) from the South Africa National Cause of Death Validation Study [22].

This should be explicitly stated, as well as any differences in how the VA may have been administered in the reference set versus the other data.

A reference on cognitive interviewing is needed. In addition, were the respondents interviewed also being interviewed for the VA? Were any changes made or requested after the cognitive interviewing when issues were identified in interpretation? As noted by a reviewer-a COREQ checklist should be completed as an appendix. Given the importance of the qualitative and emphasis on mixed methods, there remains a lack of description of the qualitative analysis (as was identified by a previous reviewer). This should be corrected and a reference for the methodology (and why chosen) added

In results, were there any differences in the 4 main areas based on either age group of the deaths (ex. Neonate versus older adults) and across countries? Any identified issues for example with linguistic or cognitive translation of questions?

How was the reference dataset used beyond the use in injury?

The discussion is much stronger after revisions. I still did not see any discussion about the reference dataset (was that for accuracy of the VA versus physician dx and if so, results and discussions?). I would only add into limitations that differences based on age group of the deceased should be included.

Reviewer #4: Nichols et al. conducted a mixed-method analysis titled, “Mixed-Methods Analysis of Selected Issues Reported in the 2016 World Health Organization Verbal Autopsy Questionnaire”, in which they show that WHO VA questionnaire requires revisions and clarifications to improve the respondents understanding of the questionnaire. In my opinion, the study can be improved by incorporating the following points:

1. The authors have not mentioned regarding how they computed the cross-tabulations data and evaluated the significance of their results, such as with a chi square test. Also mention the p value that was considered significant.

2. As the data was collected by the field team, who were included in the field teams, such as doctors or nurses? More details can be mentioned.

3. In the discussion (Line 411 – 413) when the authors have compared injury related symptom question in death due to injury vs non-injury, they should elaborate more on the reason why response was lower.

4. In WHO VA version 1.5.2 and 1.5.3, which particular respondent characteristics affect the reliability of the response? This can be mentioned to improve this part of the discussion as the authors have already highlighted that this is a sensitive issue.

5. Please add a reference for WHO VA 1.5.2/3.

6. The results state that clarity of construct was the ability to understand the terminologies and the intention of the question. However, the discussion lacks any explanations regarding the findings of clarity of construct and how this affects the questionnaire and the responses by the participants.

7. It is mentioned that an open narrative regarding the circumstances of death is also collected but it can be mentioned that what are the benefits of this? Such as can this type of questioning give more detailed and qualitative data? The authors may give more references here to support this claim.

8. The manuscript needs to be proofread for grammatical mistakes

9. The conclusion of the study needs to be improved. The authors may include particular summarized findings from the study as a properly written conclusion affects the readers’ interest.

7. PLOS authors have the option to publish the peer review history of their article (what does this mean?). If published, this will include your full peer review and any attached files.

Reviewer #1: No

Reviewer #2: No

Reviewer #4: No

---

## [Author Response · Author response to Decision Letter 1]

8 Jul 2022

The authors are grateful for the additional review and have addressed all comments as described in the Response to Reviewers.

---

## [Editor Report · Decision Letter 2]

26 Aug 2022

Mixed-Methods Analysis of Selected Issues Reported in the 2016 World Health Organization Verbal Autopsy Questionnaire

PONE-D-21-12120R2

Dear Dr. Nichols,

We’re pleased to inform you that your manuscript has been judged scientifically suitable for publication and will be formally accepted for publication once it meets all outstanding technical requirements.

Kind regards,

Prof. Ritesh G. Menezes, M.B.B.S., M.D., Diplomate N.B.

Academic Editor

PLOS ONE

---

## [Editor Report · Acceptance letter]

29 Sep 2022

PONE-D-21-12120R2 

Mixed-Methods Analysis of Select Issues Reported in the 2016 World Health Organization Verbal Autopsy Questionnaire 

Dear Dr. Nichols:

I'm pleased to inform you that your manuscript has been deemed suitable for publication in PLOS ONE. Congratulations! Your manuscript is now with our production department. 

Kind regards, 

on behalf of

Prof. Dr. Ritesh G. Menezes 

Academic Editor

PLOS ONE